# Influences of Internal Control on Enterprise Performance: Does an Information System Make a Difference?

Hani Alshaiti

Accounting Department, School of Business, King Faisal University, Al Ahsa 36363, Saudi Arabia;
hshaiti@kfu.edu.sa

**Abstract:** It is generally perceived that the effective implementation of an adequate internal control system prevents and controls an entity's risks and improves its procedures and performance. This study empirically investigates the relationship between the internal control system and firms' performance, with particular emphasis on the moderation role of an integrated information system. For this purpose, a survey was developed and sent to 215 Saudi firms that had implemented an integrated information system. A hundred and two valid responses were received. Partial least squares structural equation modeling was utilized for the data analysis and hypothesis testing. The findings confirmed that organizational structure, prospectors' strategy, information system quality, and management support significantly influence the internal control system for the study sample. The finding also supports the role of an information system as a moderator variable in the relationship between internal control and organizational performance. Additionally, the study elucidates the importance of information system maturity for information system quality.

**Keywords:** internal control system; information system; performance; contingency; enterprise resource planning system; Saudi Arabia

## 1. Introduction

Internal control (IC) has become the focus of attention whenever a scandal occurs, especially in the aftermath of the downfall of high-profit organizations such as Enron, Barings Bank, WorldCom, Tyco, and Ahold (Rae and Subramaniam 2008; Hussaini and Muhammed 2018). It has been stated that IC could be an important feature of managing an entity and defending against business failure (IFAC 2012). In other words, the IC supports the entity based on performance and is the first defensive line utilized to protect the entity's assets as well as prevent and detect its errors and fraud.

Globally, regulatory bodies recognize that the implementation of a proper IC system will normally provide an entity with reasonable assurance that it will improve the entity's performance and achieve its goals. IC can be defined as an ongoing process designed by the directors' board, chief executive officer, or other party in order to provide reasonable assurance regarding the accomplishment of the entity's objectives (COSO 1992). The definition of IC includes three levels of objectives that need to be achieved. The entity requires IC to focus on the effectiveness and efficiency of its organizational performance, productivity goals, and the safeguarding of its assets. Another objective of IC deals with the preparation of reliable financial and non-financial reports, including temporary, internal, external, and management reports. The last objective pertains to compliance with regulations and laws to which an organization is subject. IC is a vital feature in an organization's structure that can be utilized to develop and deploy monitoring across the entity to ensure sound governance (Shaiti et al. 2013). It is generally accepted that a sound IC activity can provide an organization with reasonable assurance that it can protect the entity from failure, material error, and fraud (Turnbull 1999).

Additionally, an IC system is believed to have an influence on social, environmental, and governance ratings, which indicate organization performance (Hamed 2023). An entity

with a strong IC process will outperform one with a weak process (Al-Thuneibat et al. 2015; Pakurár et al. 2019; Tao et al. 2023). Empirical evidence from Jordan shows that an IC system significantly influences financial performance and sustainability (Hamed 2023). Therefore, IC quality is an important factor that can enhance the performance of an organization (Tao et al. 2023). Although the current literature has examined the impact of an IC process on organizations' performance, the conclusions are not consistent (Hamed 2023). In addition, it is important to consider the potential influence of other factors on the IC system in order to enhance the level of IC quality. There are some prior studies that assessed the influence of different contingent factors based on different perspectives, such as (Doyle et al. 2007; Zhang et al. 2009; Abbaszadeh et al. 2019), yet more studies are needed (Doyle et al. 2007). The Doyle et al. (2007) developed a theoretical framework to provide evidence for the interaction between management support, risk management, the activities of internal audit, and the quality of internal control. Zhang et al. (2009) found an influence of financial position, organizational culture, size, management philosophy, internal auditing, and degree of decentralization on the IC system. Additionally, the authors of (Abbaszadeh et al. 2019) found that information technology (IT) was associated with IC.

In order to fill the literature gap, this study aims to investigate the influence of IC systems on organizational performance. On one hand, this study attempts to explore the influence of five contingency factors (structure, organizational culture, strategy, top management support, and information technology factor) on IC systems. On the other hand, the study further assesses the moderator impact of an information system (IS) on the relationship between IC and organizational performance. The study relationship is explained by the contingency approach, expanding on the idea that there is no single universally ideal method or approach for leading or organizing an entity. The most suitable path to take is determined by internal or external factors; thus, better organizational performance depends on achieving a more suitable alignment between a control system and these contingent variables (Fisher 1998). The study responds to a recent call by various studies (Hamed 2023; Hoai et al. 2022; Al-Muhayfith and Shaiti 2020) for further investigation to increase the understanding of the organizational and information system factors that impact IC and explain the relationship between the quality of IC and a firm's performance. By shedding light on these relationships, the study offers useful guidance to legislators and organization leaders in developing policies that enhance longstanding performance stability and growth.

The empirical results for this paper draw three conclusions. First, formalized structure, management support, strategy, and IS quality are important factors for quality IC. Second, the IC system influences organizational performance. Third, IS quality enhances the relationship between IC and organizational performance. The remainder of this paper presents the theoretical development and hypothesis formulation, research design and data collection, data analysis and findings, discussion, implementation, future research direction, and, finally, the conclusion.

## 2. Theoretical Development and Hypothesis Formulation

### 2.1. Contingency Theory Perspective

The study's theoretical framework, see Figure 1, is based on a contingency approach, which can be used as a theoretical lens to view the entity (Donaldson 2001). Fundamentally, the idea of contingency implies that something is valid only within particular circumstances (Shaiti et al. 2013). Researchers implemented contingency approaches from different aspects (Otley 1980; Felício et al. 2021; Hutahayan 2020; Abdel-Kader and Luther 2008). Otley (1980) proposed the application of the contingency theory to management accounting practices, as "there is no universally appropriate control system that applies to all organizations in all circumstances" (p. 413).

The core of the structural contingency approach model is that organizational effectiveness results from fitting organizational factors, such as organizational systems, to contingencies that reflect an organization's situation (Donaldson 2001). Rosini et al. (2020) stated

that contingency theory is used to analyze management control system (MCS) contingent fit and capability for performance. Matching the MCS to the firm's capabilities can enhance performance. Accordingly, a contingency approach can be applied to assess the influence of contingency variables and the quality of an IC system on organizational performance.

### 2.2. Formulation of Hypotheses

To locate this study within the body of existing knowledge, a brief review of the relevant literature is conducted, followed by a formulation of the eight proposed hypotheses.

#### 2.2.1. Contingency Factors and IC Quality

One main goal of this research is to examine the influence of contingency factors on the quality of an IC system. An effective IC system is important for management decision-making (Changchit et al. 2001), investors' confidence (Woods 2009; Ittonen 2010), and audit quality (Kim 2023). Thus, it is essential to review the available previous studies that investigated the contingencies that may lead to better IC. However, limited empirical studies have been conducted in the area of IC quality and its relationship with other factors (Rae and Subramaniam 2008; Doyle et al. 2007; Wang et al. 2023). Therefore, to investigate these factors, some studies related to management control systems (Felício et al. 2021) and corporate governance (Woods 2009) are referenced because IC is an integral part of them.

#### 2.2.2. Organizational Culture and Management Support

Accounting researchers have found that organizational culture and management support can affect the quality of an IC system. They argue that an organization with a high level of collaboration will have a high-quality IC system. A study conducted in China empirically examined how the characteristics of a firm could influence the quality of its IC system. The study found that organizational size, management philosophy, culture, and financial position positively impacted IC quality (Zhang et al. 2009). In addition, Sari et al. (2018) collected data from 270 Indonesian companies through a survey to examine the relationships between organizational culture, corporate governance, IC, and corporate performance. Hoai et al.'s (2022) study found that transformational leadership moderated the association between an IC system and firms' performance. These findings led us to include organizational culture, specifically collaboration, and management support as potential contingent factors in the study's model. Therefore, these relationships are examined through hypotheses H1 and H2:

**H1:** *Top management support positively impacts an IC system.*

**H2:** *Organizational culture positively influences the quality of an IC system.*

#### 2.2.3. Structure and Strategy

Considerable focus has been directed towards organizational strategy as a contingent variable of IC quality. Three types of taxonomies have been implemented in assessing the relationship between strategy and a MCS: the prospectors/analysts/defenders model (Miles and Snow 1978), the product differentiation/cost leadership classification (Porter 1980), and the build/hold/harvest model (Gupta and Govindarajan 1984). According to (Abdel-Kader and Luther 2008), these classifications are not considerably different. Consequently, the defenders/cost leaders/harvesters can reconcile at one end of the continuum, whereas the other types of strategy would be at the other end. The author of (Jokipii 2010) showed a significant relationship between organizational strategy and the quality of an IC system. A study conducted in Russia empirically examined how the organizational strategy (product differentiation) of a firm could influence its innovation and MCS dimensions. The findings indicated that three dimensions of the MCS were positively associated with the organizational strategy (Chenhall et al. 2011). In addition, the quality of an IC system can be influenced by organizational structure. Zhang et al. (2009) reported that a decentralized

organizational structure is negatively associated with the quality of IC. Verbeeten (2010) claims that business-unit structure and strategy influence control system changes. Accordingly, the following hypotheses are used to examine the relationships between structure, strategy, and IC quality:

**H3:** *Organizational structure (formalization) positively affects an IC system.*

**H4:** *There is a positive influence of prospector strategy on an IC system.*

### 2.2.4. Information System (IS)

An IS is an important factor that is reported to affect the quality of an IC system (Huang et al. 2008; Pereira et al. 2015; Valipour et al. 2012). According to Granlund (2011), "accounting researchers should ask in field and survey research a wide number of questions related to the implementation and use of Information Technology, as it may have considerable consequences regarding accounting and control practice" (p. 14). The IS concept is related to a set of components or factors that interact in order to achieve the entity's goals (Pereira et al. 2015). It can provide an entity with tools for collecting, analyzing, processing, innovating, and reporting information (Pereira et al. 2015; Monteiro and Cepêda 2021). It can also help managers develop their firms' structures, identify the important processes necessary to achieve the objectives of IC, and mitigate the entities' risks (Rubino et al. 2017). IC system researchers claim that the application of a sophisticated IS, such as the enterprise resource planning system (ERPS), is a necessary process (Huang et al. 2008; Jarah et al. 2023).

The ERPS is a new generation of IS and is normally implemented in order to integrate isolated applications into a single database (Huber et al. 2016). The integration includes data among all an entity's departments, providing the organization's manager with a broader scope. Pereira et al. (2015) illustrates that the ERP system is the one with the highest reputation in review articles. Morris (2011) showed that IS features and other characteristics can assist an organization in improving its IC system. He examined 108 United States (U.S.) firms that implemented an ERPS from 1997 to 2003 and matched them (based on industry size) with those that did not measure the impact on the IC system. Hsiung and Wang (2014) studied the effect of IT factors on IC benefits. They found information quality, system quality, and service quality influenced IC benefits. In (Abbaszadeh et al. 2019), the authors investigated the association between IT and an IC system. They collected data from auditors and managers of Iranian public entities, and they found a significant relationship between IT and an IC system. Therefore, the following hypothesis is tested:

**H5:** *The quality of an IS positively impacts the quality of an IC system.*

### 2.2.5. IS Maturity

The maturity of an IS is a crucial concept that an organization must respect when assessing IS quality (Shaiti et al. 2013). The maturity of the IS can indicate how extensively an organization understands and utilizes its system (Holland and Light 2001). It can be used to measure how well the entity's system is doing and how the entity can gain different benefits. Ragowsky et al. (2012) listed a number of factors that enable an entity to be more mature with regard to its IS, like the users of the IS, the number of constraints, and how the IS operates. Consequently, the study proposes the following hypothesis:

**H6:** *There is a positive correlation between the maturity of an IS and its quality.*

### 2.2.6. Organizational Performance

Performance can be regarded as among the most important economic concepts for an entity and for researchers. It is used to measure and evaluate how an entity works to

achieve its business objectives (Hussaini and Muhammed 2018; Akram et al. 2018; Hossin et al. 2021). Thus, it is important to focus on organizational performance. It has been argued that enhancing the quality of an IC system can drive organizational performance (Pakurár et al. 2019; Hoai et al. 2022; Wu and Zeng 2022). An effective and efficient IC system can assist an organization in fraud prevention (Handoyo and Bayunitri 2021), enhance ethical organizational behaviors, strengthen organizational mindfulness (Nguyen and Hoai 2023), and improve environmental performance (Tao et al. 2023). Al-Thuneibat et al. (2015) investigated the influence of an IC system on Saudi shareholding companies' financial performance. They found compliance with the five IC components among the sample companies. They also indicated a positive impact of IC on assets and equity returns.

Empirical evidence from North American firms found that an effective IC system supported persistent financial performance (Akisik and Gal 2017). Hussaini and Muhammed (2018) examined the influence of IC effectiveness on Nigerian banks' performance. They concluded that IC components positively impacted the banks' performance. Consistent with (Hussaini and Muhammed 2018), Pakurár et al. (2019) assessed the relationship between IC and financial performance for Jordanian banks and found that IC significantly influenced their financial performance. Rosini et al. (2020) found that management control systems directly affect performance.

Based on the financial data of China's listed companies, Wu and Zeng (2022) studied the influence of IC on enterprise performance. They found that IC positively influences organizational performance. Additionally, Hoai et al. (2022) examined the relationship between an IC system and firms' performance, the mediating role of innovation, and the moderating role of leadership. They examined 319 public Vietnamese organizations and found that an IC system influenced their organizational performance through the intensity of innovation. Hamed (2023) investigated the effect of IC systems on financial performance in Amman's listed banks. They found that the study sample's banks complied with IC constraints and that IC system compliance significantly influenced their financial performance and sustainability. Consequently, the author proposes the following hypothesis:

**H7:** *IC quality positively affects organizational performance.*

### 2.2.7. The Moderating Role of IS Quality

An IS is basically a set of components used to collect, process, store, and analyze data and to provide relevant information to support management operations and decisions (Hailu 2014). An IS plays a significant role when it influences a firm's functions, performance, and productivity. According to Kehinde and Soyebo (2012), a good quality IS enables an organization to prepare an effective plan, control the organization's activities, and make more accurate decisions. A study (Hailu 2014) found that an IS impacts organizational performance. Additionally, Al-Muhayfith and Shaiti (2020) found that an integrated IS, such as SAP and Oracle, influenced organizational performance. Therefore, this study argues that the impact of IC quality on organizational performance would be enhanced by greater IS quality, and it thus proposes the following hypothesis:

**H8:** *The quality of an IS moderates the relationship between the quality of IC and business performance.*

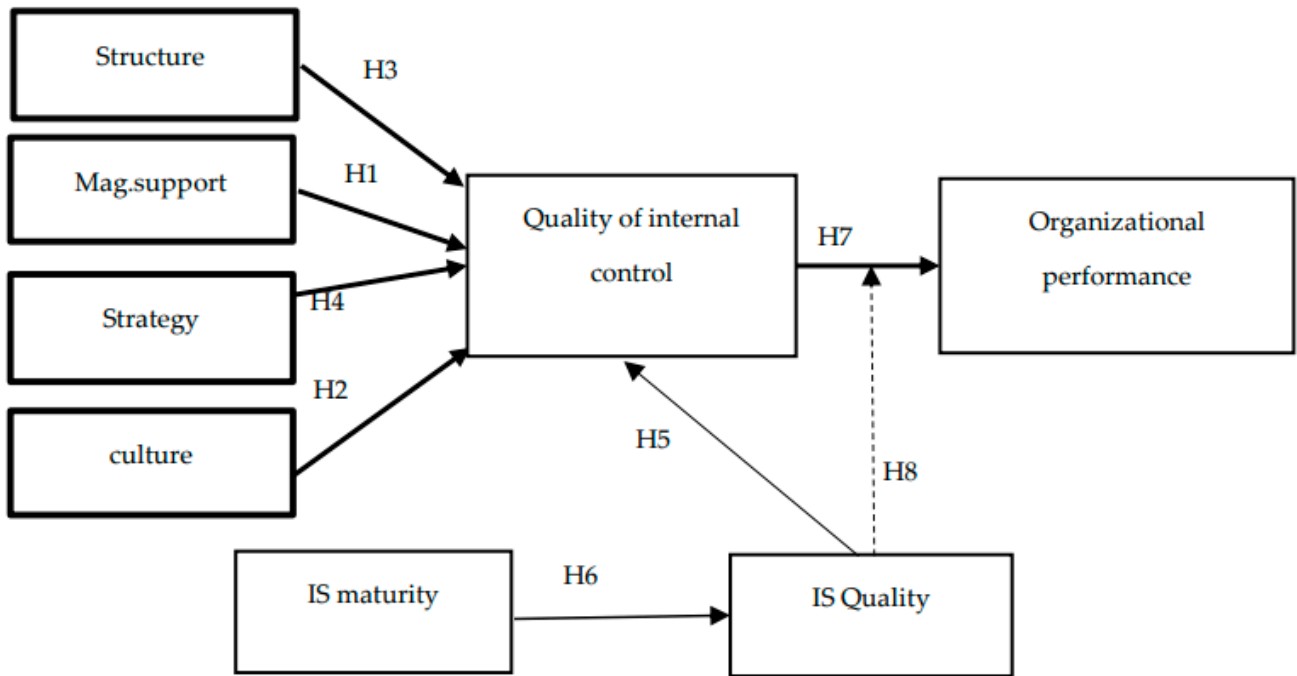

**Figure 1.** The theoretical framework.

## 3. Research Design and Data Collection

### 3.1. Research Method

This study draws on the principle of contingency theory in accounting studies to develop a contingency model of IC system organizational performance, which could help to understand, describe, and predict the quality of an IC system in relation to organizational performance. In line with the study's theoretical framework, a positive approach is adopted, which involves collecting quantitative data to address the research questions (Abdel-Kader and Luther 2008). This approach relies on the presumption that social reality is relatively objective, and it relies on empirical evidence (Saunders et al. 2009).

For this study's aim and objectives, the positivist approach is applicable as it empirically develops a theoretical framework for effective corporate performance. The positivist approach is aligned with the deductive model. Thus, the study began by building the theoretical framework, which contains eight hypotheses, after reviewing the literature. A survey strategy, particularly the questionnaire technique, was applied to collect the main data and assess the hypotheses. This method is commonly applied for theory testing within the management accounting discipline (Abdel-Kader and Luther 2008) because it facilitates the collection of a large amount of data from a considerable population and helps to control cost and time constraints (Saunders et al. 2009). Structural equation modeling (SEM) is applied for analyzing the data in this study. SEM is one of the sophisticated statistical techniques utilized by a number of researchers to promote theory development and model testing (Hair et al. 2013; Akram et al. 2018). In SEM, there are two approaches that can be applied to estimate study relationships: covariant-bases SEM (CB-SEM) and partial least squares SEM (PLS-SEM). For this study, the PLS-SEM is implemented because it has the minimum demands on residual distributions, measurement scales, and a large sample size (Hair et al. 2013).

### 3.2. Sample and Data Collection

All Saudi companies that were found to have implemented an integrated IS, such as SAP, Oracle, and PeopleSoft systems, were included in the study sample. However,

since there was little data on Saudi firms that had implemented an integrated IS in the previous literature, different data sources were utilized in the present study. These included several experimental studies that were conducted on Saudi businesses, like (Al-Muharfi 2010; Al-Turki 2011; Al-Muhayfith and Shaiti 2020). Some of the ERPS vendors in Saudi Arabia were contacted to obtain the names of the companies that had implemented an ERP system in Saudi Arabia (as ERP is an integrated IS). Moreover, some websites were used, such as the Ministry of Commerce and Industry. A list of 215 companies that had implemented an integrated IS in Saudi Arabia was obtained. Finally, emails were sent to the 215 companies in order to confirm that the companies had an integrated IS and were willing to participate in the study.

For the data collection, an initial questionnaire was developed based on the literature review, and it was reviewed by three academic staff and two professionals. In addition, the ethical committee at King Faisal University (the sponsor body) confirmed that there were no ethical issues related to the questionnaire items. The questionnaire was distributed among the participants by email, and 108 valid responses were received (a 50% response rate). After collecting the empirical data, the developed model and its hypotheses were assessed using SmartPLS 4 software.

*3.3. Research Instruments*

A survey with the following structure was developed and used to measure the study's eight constructs: The study's main aim and the confidentiality of the provided information were stated on the first page of the survey to avoid bias in the responses. The questionnaire consisted of four sections. The first one assessed the quality of the IC system; the four objectives of the Committee of Sponsoring Organizations (COSO) farmwork were used. The following section measured business performance with four indicators (economic, financial, environmental, and communication). The third section included all the statements that measured the study's contingencies: structure, strategy, management support, organizational culture, IS, and IS maturity. The last section included questions that related to demographic data (IS brand, company type, and participants' education). The items used to measure the constructs were selected based on previous research (see Table 1). The five-point Likert scale was applied with 'Strongly Disagree', for one point, and 'Strongly Agree', for five points.

**Table 1.** Questionnaire items.

| Construct | Number of Measures | Source |
|---|---|---|
| 1. IS maturity | 5 items | (Nolan 1979) |
| 2. Culture (collaboration) | 3 items | (Detert et al. 2000; Jones et al. 2006) |
| 3. Top management support | 5 items | (Jaworski and Kohli 1993; Nahavandi and Malekzadeh 1993; Linying et al. 2009) |
| 4. Strategy (prospector) | 5 items | (Croteau and Bergeron 2001; Felício et al. 2021) |
| 5. Structure (formalization) | 3items | (Pugh et al. 1968; Donaldson 2001) |
| 6. IS quality | 7 items | (DeLone and McLean 1992; Myers et al. 1997; Gable et al. 2003) |
| 7. Internal control quality | 4 items | (COSO 2004) |
| 8. Performance | 4 items | (Staniškis and Arbačiauskas 2009; Akram et al. 2018; Al-Muhayfith and Shaiti 2020) |

Additionally, the PLS-SEM approach was applied for data analysis. PLS is a structure equation modeling technique that provides a promising avenue for a latent variable path modeling technique to be applied without being constrained by some presumptions, like normality or the need for a substantial sample size (Fornell and Larcker 1981). Many researchers have utilized this approach in relation to management support (Fornell and Larcker 1981), information systems (Vinzi et al. 2010), and accounting research (Hall and Smith 2009; Lee et al. 2011; Akram et al. 2018; Widyastuti et al. 2023).

## 4. Data Analysis and Findings

In the first stage of the data analysis, Statistical Product and Service Solutions (SPSS) 26 software was used to process descriptive statistics to assess the respondents' demographic characteristics (firm type, qualification, and IT brand). For the firm types listed in the questionnaire, the highest response rate came from joint liability firms (39.5%), followed by public firms (30%), whereas the lowest response rate came from private firms (6%). This implies that applying an integrated IS is not common among private firms. This is because the majority of private firms are small, and it would be costly to implement a high-quality system.

The results in Table 2 show that most of the participants have a bachelor's degree. Approximately 74% of the respondents have a bachelor's degree in accounting, finance, or management, whereas 26% have a diploma degree. Thus, the respondents were relatively knowledgeable in the research area. Moreover, the results show that Saudi enterprises employ over 30 different brands of integrated IS software. According to Al-Muhayfith and Shaiti (2020), integrated IS software can be classified by well-known systems, such as SAP, Microsoft Dynamic AX, and Oracle, and less well-known systems, such as Peachtree, Solution, and HeloolERP. Table 2 shows that 74% of the respondents used well-known IS software, whereas only 26% used less well-known software.

**Table 2.** Demographic characteristics of the respondent companies.

| Characteristics | | N | n% |
|---|---|---|---|
| Firm Type | Private firm | 6 | 5.5% |
| | Limited partnership | 27 | 25% |
| | Joint liability firm | 42 | 39.5% |
| | Public firm | 33 | 30% |
| Qualification | Bachelor in Acc. or Fin. | 69 | 64% |
| | Bachelor in Bus. Man. | 10 | 10% |
| | Diploma in Acc. or IS | 29 | 26% |
| IT Brand | Well-known | 80 | 74% |
| | Less well-known | 28 | 26% |

In the next stage, the PLS-SEM approach was utilized to examine the correlation coefficient and assess the influences between the study's variables (Lee et al. 2011). Two PLS-SEM models were used for data analysis (Hair et al. 2013). The first one is the outer model, which identified and assessed the correlation between the observation data and the latent variables. Particularly, it assessed internal consistency reliability, composite reliability, and discriminate and convergent validity. The second model was the inner model, which was used to assess the relationships between the study variables using a structure scale (Hair et al. 2013).

### 4.1. The Outer Model

Before assessing the significance of the relationships among the study's constructs, it is important to evaluate the outer or measurement model. First, the study removed all cases that had over 10% of missing data. Thus, six cases were removed. Second, the factor loading for each indicator was examined. All items loaded onto their respective constructs; however, eight items (ISQ4, ISQ5, Strategy 4, MS2, MS4, Maturity 4, Structure 3, and Performance 1) had factor loadings of less than 0.5, which added a few values to the model explanation; thus, these were removed (Hulland 1999; Hair et al. 2010). Table 3 shows the factor loadings of the indicators after removing low-loading factors.

**Table 3.** Validity and reliability of the constructs.

| Construct Item | Loading | CA | CR | AVE | R² |
|---|---|---|---|---|---|
| **Management Support** | | 0.758 | 0.857 | 0.688 | |
| -Supporting research, development, and innovation | 0.692 | | | | |
| -Preferring to delegate tasks to others | 0.835 | | | | |
| -Staff are involved in strategic planning and technical direction | 0.911 | | | | |
| **Culture (Collaboration)** | | 0.754 | 0.860 | 0.674 | |
| -Usually, staff choose to work in project teams | 0.864 | | | | |
| -Staff are willing to share data between them | 0.895 | | | | |
| -Entity inspires its staff to be free in their thinking and in producing ways to do their teamwork-related jobs | 0.690 | | | | |
| **Strategy** | | 0.877 | 0.916 | 0.732 | |
| -Development of new products and services is supported by the entity's mission and actions | 0.778 | | | | |
| -Entity leads its sector in the direction of innovation | 0.856 | | | | |
| -Usually, the entity's actions lead to new rounds of competitive advantage in the sector | 0.892 | | | | |
| -Entity is generally involved in high-risk projects | 0.890 | | | | |
| **Structure (Formalization)** | | 0.762 | 0.892 | 0.805 | |
| -Entity has diversified occupational specialties | 0.869 | | | | |
| -Description of the job is presented | 0.925 | | | | |
| **IS Maturity** | | 0.827 | 0.882 | 0.653 | |
| -Number of IS users has increased since implementation | 0.875 | | | | |
| -IS applications that are used satisfy the entity's needs | 0.833 | | | | |
| -Satisfaction with the control processes and standards of the IS's resources | 0.760 | | | | |
| -Since the implementation of the IS, the control of traditional data processing activities has become tighter | 0.759 | | | | |
| **IS Quality** | | 0.843 | 0.889 | 0.617 | 0.29 |
| -Entity's [IS] is easy to use | 0.861 | | | | |
| -Entity's [IS] is easy to learn | 0.805 | | | | |
| -Entity's [IS] meets the users' requirements | 0.808 | | | | |
| -Entity's [IS] requires a minimum number of computers | 0.656 | | | | |
| -Data within the [IS] are fully integrated | 0.784 | | | | |
| **Internal Control Quality** | | 0.803 | 0.871 | 0.627 | 0.57 |
| -Compliance with applicable laws and regulations | 0.771 | | | | |
| -Effectiveness and efficiency of operations | 0.858 | | | | |
| -Reliability of financial reporting | 0.793 | | | | |
| -Management efficiently establishes its strategic objectives | 0.743 | | | | |
| **Performance** | | 0.652 | 0.811 | 0.590 | 0.39 |
| -Entity responds quickly to the first signs of opportunity/innovation | 0.699 | | | | |
| -Normally, profitability targets are achieved | 0.770 | | | | |
| -Operation and product characteristics are improving | 0.830 | | | | |

Third, Cronbach's alpha, composite reliability, and convergent validity were assessed. As shown in Table 3, based on the value of Cronbach's alpha, the composite reliability for all of the constructs is above 0.7 (Hulland 1999; Hair et al. 2010), except for performance, whose value is less than the established criterion of 0.70. According to Hair et al. (2013) and Nunnally (1978), researchers generally consider the minimum level to be 0.70; however, lower correlations for a questionnaire item might be justifiable and acceptable. A benchmark for Cronbach's alpha value of 0.60 and above is common in the context of IS and technology studies (Mano et al. 2023; Yohans et al. 2023).

Fourth, the average variance extracted (AVE) values for all the constructs are more than 0.5 degrees, which establishes convergent validity (Fornell and Larcker 1981). Fifth, the discriminant validity was assessed utilizing the Fornell-Larcker technique (Fornell and

Larcker 1981). Table 4 shows an agreeable level of discriminant validity, as the square roots of the AVE values are greater than the inter-latent construct correlation.

**Table 4.** Correlation matrix among the study constructs (AVE square root).

|  | Culture | Formalization | ICQ | ISQ | MS | Maturity | Perform. | Strategy |
|---|---|---|---|---|---|---|---|---|
| Culture | *0.821* |  |  |  |  |  |  |  |
| Formalization | 0.441 | *0.897* |  |  |  |  |  |  |
| ICQ | 0.352 | 0.582 | *0.792* |  |  |  |  |  |
| ISQ | 0.200 | 0.225 | 0.479 | *0.786* |  |  |  |  |
| MS | 0.432 | 0.350 | 0.488 | 0.125 | *0.818* |  |  |  |
| Maturity | 0.223 | 0.240 | 0.467 | 0.546 | 0.210 | *0.808* |  |  |
| Perform. | 0.408 | 0.417 | 0.606 | 0.347 | 0.551 | 0.394 | *0.768* |  |
| Strategy | 0.422 | 0.638 | 0.628 | 0.329 | 0.514 | 0.281 | 0.500 | *0.855* |

Note: Values in italics represent the square root of the AVE value.

### 4.2. Testing the Hypotheses

After satisfactorily analyzing the outer model, the next step in PLS-SEM analysis is to assess the inner model. Figure 2 was estimated using the SmartPLS 4 software to assess the inner model and test the hypotheses. One main standard assessment criterion is the coefficient of determination ($R^2$) technique. As in ordinary least squares regression (OLS), the $R^2$ value in PLS illustrates the proportion of the total variance of the variables that is described by the model (Hair et al. 2013). The $R^2$ of the endogenous variable is the predictive power utilized to assess the inner model. It is typically the first value evaluated by researchers (Chin 2010). Table 3 shows that the value of $R^2$ for the IC quality is 0.572, which means that formalization, management support culture, strategy, and IS quality explain 57.2% of the variance in the IC quality. In addition, the $R^2$ for organizational performance is 0.397, which means that IC quality and the indirect effect of the study's organizational factors explain 39.7% of the variance in organizational performance. Compared to other studies in the accounting field, such as those by (Hartmann 2005; Naranjo-Gil and Hartmann 2006; Chenhall et al. 2011; Al-Muhayfith and Shaiti 2020; Hoai et al. 2022), these $R^2$ values fall within an acceptable range.

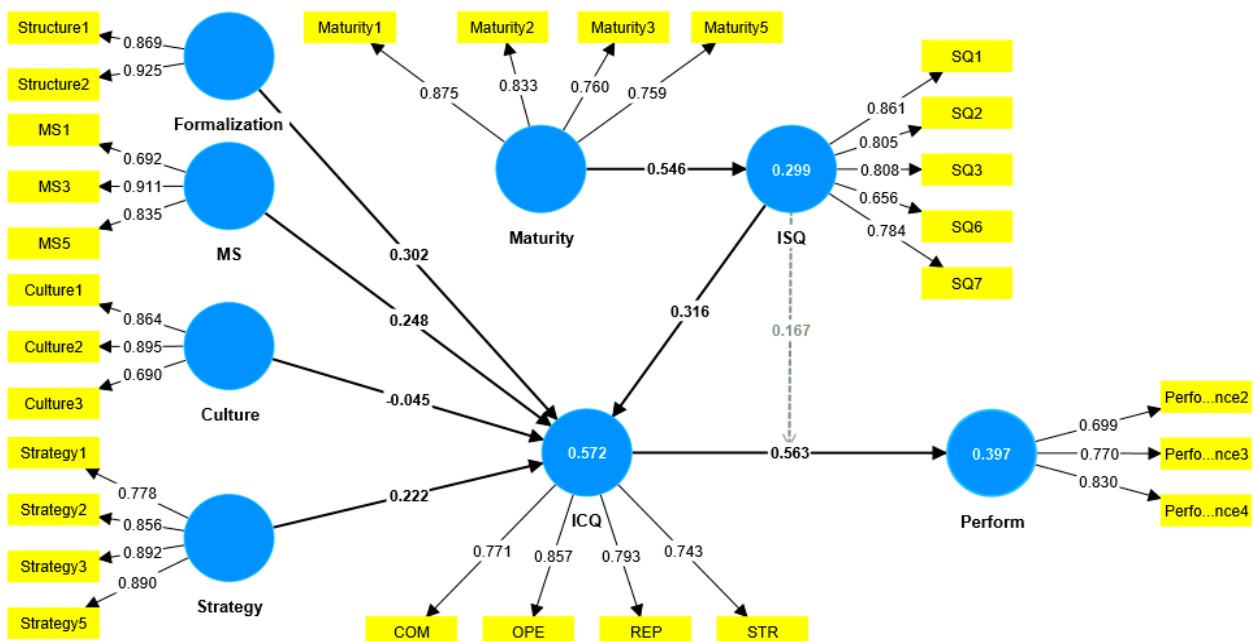

**Figure 2.** Path coefficients with a path diagram for the whole sample.

Moreover, the predictive relevance technique is used in addition to the path coefficient ($R^2$) technique to evaluate the structure model (Chin 2010). This technique assesses the model's capability to predict. The predictive relevance ($Q^2$) results hypothesize whether the model is capable of predicting each endogenous latent variable's indicators adequately (Hair et al. 2013). For the study, $Q^2$ was assessed using the cross-validity communality, as illustrated in Table 5, which should be greater than zero in order to indicate that the model is considered to have predictive validity.

**Table 5.** Cross-validity communality.

| Construct | Cross-Validity Communality |
|---|---|
| 1. IS maturity | 0.69 |
| 2. Culture (collaboration) | 0.87 |
| 3. Top management support | 0.76 |
| 4. Strategy (prospector) | 0.74 |
| 5. Structure (formalization) | 0.78 |
| 6. IS quality | 0.73 |
| 7. Internal control quality | 0.61 |
| 8. Performance | 0.65 |

Table 5 shows positive $Q^2$ results for the eight constructs, which suggests the study's model has predictive relevance.

In addition, the software makes no distributional assumptions; thus, bootstrapping (5000 samples with replacement) was applied to test the statistical significance of each path coefficient (Hair et al. 2013). Table 6 presents the results for the significance of each path coefficient. Among the contingency variables, management support, organizational strategy, and formalization were found to be significantly associated with IC quality ($\beta$ = 0.248, $\rho$ = 0.000; $\beta$ = 0.222, $\rho$ = 0.015; and $\beta$ = 0.302, $\rho$ = 0.001, respectively) at a level of 0.5. Additionally, a positive and significant relationship between IS quality and IC quality was found ($\beta$ = 0.316, $\rho$ = 0.000). However, organizational culture showed no significant effect on IC quality ($\beta$ = -0.045, $\rho$ = 0.555). Furthermore, IS maturity was significantly related to IS quality ($\beta$ = 0.546, $\rho$ = 0.000). The results also showed that IC quality significantly influenced organizational performance ($\beta$ = 0.563, $\rho$ = 0.000).

**Table 6.** Significance of the path coefficients.

| | Beta | Standard Deviation (STDEV) | T Statistics | *p* Values | Decision |
|---|---|---|---|---|---|
| Culture → ICQ | −0.045 | 0.077 | 0.274 | 0.555 | Non-Sig |
| Formalization → ICQ | 0.302 | 0.093 | 3.260 | 0.001 | Sig |
| MS → ICQ | 0.248 | 0.068 | 2.627 | 0.000 | Sig |
| Maturity → ISQ | 0.546 | 0.088 | 6.178 | 0.000 | Sig |
| Strategy → ICQ | 0.222 | 0.091 | 2.433 | 0.015 | Sig |
| ISQ → ICQ | 0.316 | 0.088 | 3.606 | 0.000 | Sig |
| ICQ → Perform. | 0.563 | 0.075 | 7.437 | 0.000 | Sig |
| ISQ × ICQ → Perform. | 0.167 | 0.083 | 2.026 | 0.043 | Sig |

The study also hypothesized that IS quality would have a moderating influence on the relationship between IC quality and organizational performance. The moderation analysis was calculated by utilizing the PLS product indicator method. As stated by Becker et al. (2018), in a structural model, a moderator factor can influence the strength or even the direction of the relationship between constructs. Therefore, to assess the possibility of IS quality having a moderating effect, IC quality as a predictor and IS quality as a moderator were multiplied to create an interaction construct and predict organizational performance. Table 6 indicates that the estimated standardized path coefficients for the influence of IC quality × IS quality on organizational performance ($\beta$ = 0.167; $\rho$ = 0.043) were positive

and significant at a level of 0.05. This indicates that the positive influence of IC quality on business performance is stronger when IS quality is at a high level, as shown in Figure 3. Therefore, H8 is supported.

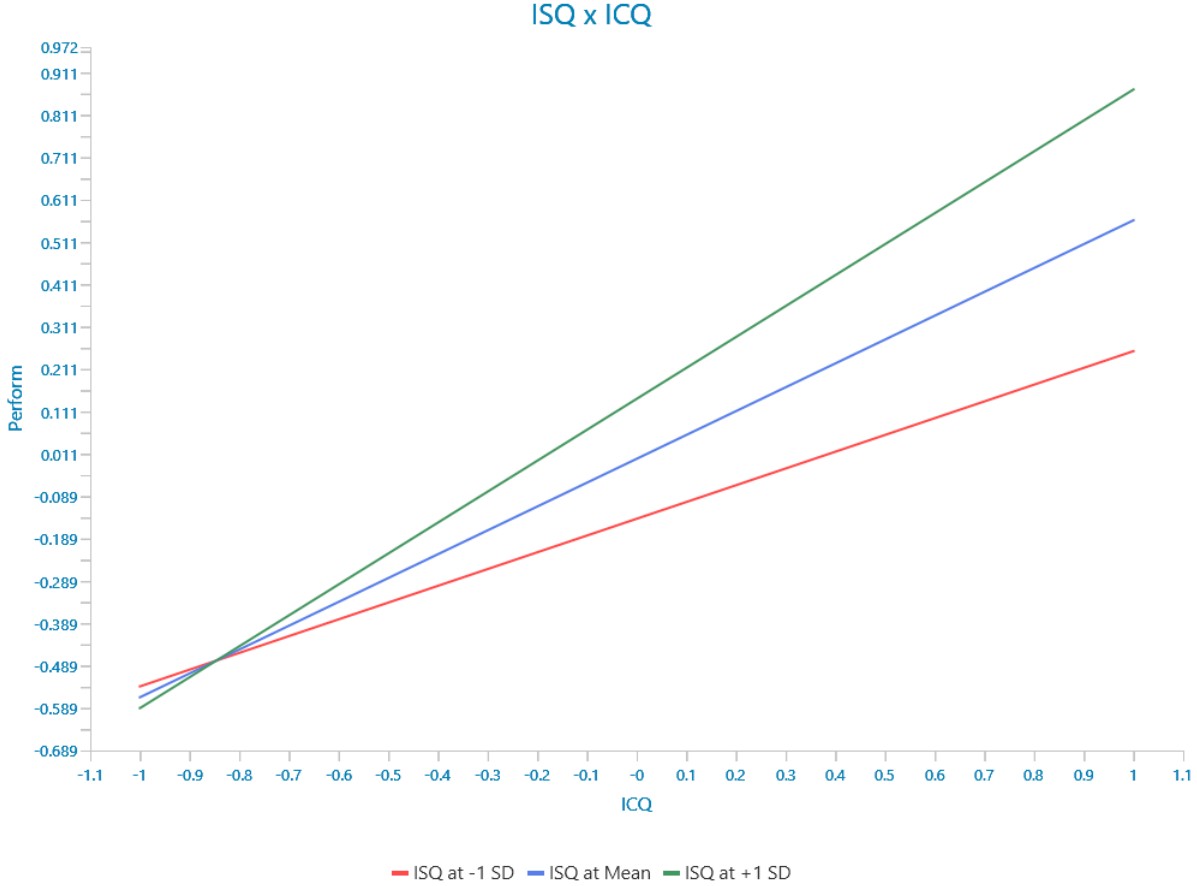

**Figure 3.** Moderator effect.

## 5. Discussion

This study developed and validated a theoretical model based on empirical data gathered within the context of the Saudi Arabian corporate environment. In particular, the study examined eight hypotheses. A regression analysis was conducted to assess the study hypotheses. Based on the findings shown in Table 6, seven hypotheses were supported, and one was rejected. Additionally, the adjusted $R^2$ was 0.572 for IC quality and 0.397 for organizational performance, which fall within an acceptable range compared with those in many previous studies in the accounting field (Al-Muhayfith and Shaiti 2020; Hoai et al. 2022; Hamed 2023).

The findings of hypothesis testing support the proposed correlation between management support (assessed by three indicators) and IC quality in Hypothesis 1. The current study is consistent with those of (Abernethy et al. 2010; Doeleman et al. 2012). Both studies indicated that leadership style significantly predicts the control system. According to Jarah et al. (2023), an IC system is a process influenced by an entity's managers that provides sufficient confidence about the accomplishment of the entity's objectives. Surprisingly, the study's results show no evidence of the impact of organizational culture (collaboration) on IC quality. This means that staff believe that individual work would be more valuable for the quality of IC than teamwork. This indicates that a high-quality IC system may be based on an individualistic workplace culture. This is consistent with Alzeban (2015), which indicated that individualism positively impacted the quality of internal audits.

For organizational structure, the findings showed a positive influence of formalization on IC quality. These findings are consistent with those obtained in contingency studies,

which indicate that formalized and specialized structures are correlated with a sound MCS (Hutahayan 2020; Felício et al. 2021). Dropulić and Rogošić (2014) examined the term of an MCS as a package of controls and indicated that most of Croatia's firms used formalized management control systems. Additionally, Shafie et al. (2019) found that organizational structure significantly impacted the effectiveness of IC.

Furthermore, for Hypothesis 4, the finding supports the authors, who argued that the acceptance of a prospector strategy would positively correlate with the field of MAS (Abdel-Kader and Luther 2008; Verbeeten 2010; Rahi et al. 2022), which means that the sample firms should concentrate on features such as innovation, competitive advantage, and strategic planning. The findings are also consistent with the results of (Jokipii 2010; Chenhall et al. 2011). Jokipii (2010) showed that strategy was positively correlated with control. In more detail, a study conducted in Russia empirically examined how the organizational strategy (product differentiation) of a firm could influence the MCS dimensions. Three dimensions of the MCS were used: formal controls, organic innovative culture, and a package of controls composed of social networking. The findings indicated that strategy and product differentiation were positively associated with the three MCS dimensions (Chenhall et al. 2011).

According to the results of the structural model, the quality of an IS positively influences that of IC. The finding is consistent with those in some studies, such as (Morris 2011; Valipour et al. 2012; Monteiro and Cepêda 2021; Jarah et al. 2023). Studies such as Morris (2011) document that IC weaknesses are reported less for companies that use IT systems than those who do not. These studies specify compelling proof of how an IS plays a crucial role in enhancing the quality of IC. Using qualitative data, Valipour et al. (2012) showed that the five COSO's framework components were affected by the adoption of the ERPS. In addition, Jarah et al. (2023) found that an accounting IS, specifically system and information quality, significantly influenced an IC system by providing several methods and processes to facilitate it.

In addition, the empirical results showed a positive association between IS maturity and IS quality. This result aligns with the idea that the level of maturity could influence IS quality in varying ways (Ragowsky et al. 2012; Al MohamadSaleh and Alzahrani 2023). The authors of Voordijk et al. (2003) found that the successful implementation of the ERPS depended on the strategic role of IT, IS maturity, firm strategies, and the implementation approach. Furthermore, Dias and Souza (2004) illustrated that the ERPS maturity stage can be recognized as a competitive advantage for an entity. Additionally, Suh et al. (2017) found a moderating influence of IS maturity on the association between IS investment and the success of IS quality. They indicated that even with the existence of a robust impact of IS investment on IS quality, IS maturity, as a moderator, improved the impact of IS investment on IS success.

The main purpose of this study was to examine the impact of IC quality on organizational performance. It has been stated that IC is a fundamental feature for preventing risks and a prerequisite for successful operations and performance (Rae and Subramaniam 2008). The current study's findings support this argument. The results show that IC quality strongly influences organizational performance ($\beta = 0.563$, $\rho = 0.000$). This result is consistent with findings in previous empirical studies (Hussaini and Muhammed 2018; Pakurár et al. 2019; Felício et al. 2021; Hoai et al. 2022; Hamed 2023). Pakurár et al. (2019) examined the impacts of supply-chain integration and IC on Jordanian banks' financial performance and found them significant. They illustrated that IC is positively associated with financial performance. Consistently, Hoai et al. (2022) reported that IC effectiveness positively influenced firms' performance. Hamed (2023) examined the effect of an IC system on the banking sector's financial performance. It indicated that an effective IC system could be a key factor for appropriate performance.

Finally, the finding related to Hypothesis 8 indicated that IS quality moderates the relationship between IC quality and organizational performance ($\beta = 0.167$; $\rho = 0.043$). The results imply that even with the existence of a strong effect of IC quality on performance,

IS quality as a moderator improves the impact of IC quality on firms' performance. No previous study has illustrated this relationship. However, the findings in (Hailu 2014; Al-Muhayfith and Shaiti 2020) partially support the current study findings. Both studies found that an IS (ERPS) influenced organizational performance.

## 6. Implications and Future Research Directions

Although previous studies have examined the link between an IC system and organizational performance (Akisik and Gal 2017; Hussaini and Muhammed 2018; Pakurár et al. 2019; Hoai et al. 2022; Hamed 2023), this study could be the first to investigate the influence of IS quality as a moderator variable on the relationship between IC quality and firm performance. Additionally, the study's findings emphasize the important role of an IC system and other contingency variables in enhancing organizational performance. Practically, this study has implications for the adoption of an IC system and its influence on organizational performance. Company managers must not only implement effective IC systems but must also consider some important factors, such as company strategy and structure, in implementing an effective IS to enhance their companies' performance. Additionally, the findings urge companies without an IS or those planning to implement one to recognize the importance of implementing an effective IS.

This study has some limitations that justify the need for additional discussion and the opportunity to explore future research avenues. From a theoretical viewpoint, because of the complexity of the study's structure model, the author could not identify and include all potential organizational factors that may influence the two main constructors. According to Al-Thuneibat et al. (2015), the influence of an IC system on organizational performance may vary depending on different organizational factors. Future research can explicitly include different factors, for example, some leadership attributes, politics, uncertainty, competition, sustainability practices (Al-Thuneibat et al. 2015; Hoai et al. 2022; Rahi et al. 2022), and other organizational factors, to address their effects on the study's main relationships.

From an empirical viewpoint, this study's context is confined to a specific subset of Saudi entities that have implemented an IS. Future research could be enhanced by conducting research on two types of firms: IS-implemented firms and non-IS-implemented firms. This would help to compare and provide a better understanding of the relationships between the study's variables and enhance the generalizability of the findings.

## 7. Conclusions

Based on the importance of the IC system and its role in enhancing the performance of an organization, this study developed a structural model of the contingency variables and IC quality that could predict an improvement in organizational performance for Saudi firms. This is a topic that has not yet been explored adequately in the literature. Therefore, the structural model was first developed, followed by an assessment of the variables that predict organizational performance. The empirical results of the structure model indicated several noteworthy observations. Firstly, the findings provided evidence that supports the impact of IC systems on organizational performance. Secondly, the results showed the effectiveness of management support, IS quality, organizational structure, and strategy on the quality of an IC system. These findings prove that top management is more interested in having a high level of IC quality. Additionally, a formalized structure provides an organization with an adequate IC system. The findings also offer evidence that supports the importance of an IS in enhancing the quality of IC. Lastly, the study illustrated the moderating effect of IS quality on the relationship between an IC system and organizational performance.

**Funding:** The authors acknowledge the Deanship of Scientific Research at King Faisal University for financial support (Grant No. 4034).

**Institutional Review Board Statement:** The study was conducted according to the guidelines of the Declaration of Helsinki and approved by the Deanship of Scientific Research Ethical Committee, King Faisal University (project number: GRANT4034, date of approval: 1 May 2023).

**Informed Consent Statement:** Informed consent was obtained from all subjects involved in the study.

**Data Availability Statement:** Data can be made available upon request.

**Conflicts of Interest:** The author declares no conflict of interest.

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
