# Peer review of "Influences of Internal Control on Enterprise Performance: Does an Information System Make a Difference?"

_jrfm, doi:10.3390/jrfm16120518_

Round 1
Reviewer 1 Report
Comments and Suggestions for Authors
This paper analyzed the influence of IT-based solution, namely information systems, in organization's internal control. The author gives a theoretical framework consisting of eight hypotheses to observe the influences of IS on IC. After a survay was completed with over a hundred responses, the results were shown in statistical format.
First of all, the paper is well written and is easy to follow even for a non-expert in this field. Secondly, the paper is well organized and the structure is clearly shown.
This reviewer has several minor questions that the author can address in a revision of the paper:
1. Table 4 shows the correlation matrix between the study constructs. Please include the significance factor in the matrix (p values) in order to understand the results more clearly. ,
2. Figure 3 is not scaled well and the text on the axis is skewed. Please include a clearer figure in a higher resolution.
Overall, this reviewer thinks that this paper should be accepted after minor revisions.
Author Response
Comments 1: This paper analyzed the influence of IT-based solution, namely information systems, in organization's internal control. The author gives a theoretical framework consisting of eight hypotheses to observe the influences of IS on IC. After a survey was completed with over a hundred responses, the results were shown in statistical format. First of all, the paper is well written and is easy to follow even for a non-expert in this field. Secondly, the paper is well organized and the structure is clearly shown |
||
Response 1: Thank a lot for your comment and I really appreciating your evaluation.
|
||
Comments 2: Table 4 shows the correlation matrix between the study constructs. Please include the significance factor in the matrix (p values) in order to understand the results more clearly. |
||
Response 2: Thank you for pointing this out. The P value of each path coefficient is presented in table 5. Additionally, the study assesses the discriminant validity by using the Fornell-Larcker technique. The Fornell-Larcker method requires the AVE (the average variance extracted) for each construct to be larger than its squared inter-correlation with other constructs. The constructs correlation matrix was developed automatically by the SmartPLS software (as show in table 4) in order to assess the discriminant validity through the Fornell-Larcker technique. The table shows that the all square roots of AVE for the study constructs are higher than the correlation between each construct and another (in same row or column). Therefore, all the constructs have an acceptable level of discriminant validity for the study. A note has been added in line 350. Please see page number 9 & 11.
|
||
4. Response to Comments on the Quality of English Language |
||
Point 1: English language fine. No issues detected |
||
Response 1: Thank you. |
||
5. Additional clarifications |
||
I have reviewed the introduction, methodology and results. Also, I have added some recent references and. Please see the is change, I have highlighted that in the re-submitted files. In addition I have delated a number of old bibliography which are not important to the study finding such as: -Collis, J.; Hussey, R. Business Research, A Practional Guide for Undergraduate & Postgraduate Students, 3rd ed.; Palgrave Macmillan: New York, 2009. -Langfield-Smith, K. Management Control Systems and Strategy: A Critical Review. Accounting, Organizations and Society 1997, 22, 207-232. -Anderson, S. W.; Young, S. M. The Impact of Contextual and Process Factors on the Evaluation of Activity-Based Costing Systems. Accounting, Organizations & Society 1999, 24, 525-559. -Otley, D. Performance management: a framework for management control systems research. Management Accounting Research 1999, 10, 363-382. -Chang, S.I.; Jan, D. SOX 404-compliant ERP System Internal Control Framework - The Preliminary Outcome. Journal of Business and Policy Research 2010, 5, 282-295. -Klamm, B. K.; Watson, M. W. Sox 404 Reporting Internal Control Weaknesses: A Test of COSO Framework Components and Information Technology. Journal of information systems 2009, 23, 1-23
|

Reviewer 2 Report
Comments and Suggestions for Authors
1. The measurement of the eight constructs is not sufficiently explained.
2. The results of the tested hypotheses are not sufficiently explained.
3. There are certain references errors such as 91. Dropulić, I.; Rogošić, A. Formalization of management control systems ......
Author Response
Comments 1: The measurement of the eight constructs is not sufficiently explained. |
||
Response 1: Thank you a lot for your comment, the measurement items for each constructs have been added in table 3 to emphasize this point., please see page 8 and line 335
|
||
Comments 2: The results of the tested hypotheses are not sufficiently explained. |
||
Response 2: I agree. I have, accordingly, explained the result of the tested hypotheses section more and I have added another technique for explaining the structure model with a new table ( Table 5, page 10 and line 373) in order to emphasize this point., please see the page 10 and lines 355-357 & 366-373 & 375-376
|
||
4. Additional clarifications |
||
I have reviewed the introduction, methodology and results. Also, I have added some recent references and. Please see the is change, I have highlighted that in the re-submitted files. In addition I have delated a number of old bibliography which are not important to the study finding such as: -Collis, J.; Hussey, R. Business Research, A Practional Guide for Undergraduate & Postgraduate Students, 3rd ed.; Palgrave Macmillan: New York, 2009. -Langfield-Smith, K. Management Control Systems and Strategy: A Critical Review. Accounting, Organizations and Society 1997, 22, 207-232. -Anderson, S. W.; Young, S. M. The Impact of Contextual and Process Factors on the Evaluation of Activity-Based Costing Systems. Accounting, Organizations & Society 1999, 24, 525-559. -Otley, D. Performance management: a framework for management control systems research. Management Accounting Research 1999, 10, 363-382. -Chang, S.I.; Jan, D. SOX 404-compliant ERP System Internal Control Framework - The Preliminary Outcome. Journal of Business and Policy Research 2010, 5, 282-295. -Klamm, B. K.; Watson, M. W. Sox 404 Reporting Internal Control Weaknesses: A Test of COSO Framework Components and Information Technology. Journal of information systems 2009, 23, 1-23
|

Reviewer 3 Report
Comments and Suggestions for Authors
1) very old bibliography, should focus on recent articles 2021, 2022 and 2023
2) the introduction must have a description of the article's sections. the purpose of the article must be better defined
3) to complement items 2.2.4 and 2.2.5 I suggest the article
PEREIRA, FERNANDA DE CARVALHO ; VEROCAI, HENRIQUE DONDEO ; CORDEIRO, VINÍCIUS RIBEIRO ; GOMES, CARLOS FRANCISCO SIMÕES ; COSTA, HELDER GOMES . Bibliometric Analysis of Information Systems Related to Innovation. Procedia Computer Science, v. 55, p. 298-307, 2015. https://doi.org/10.1016/j.procs.2015.07.052
4) item 3.1 must clearly report the methodology to be applied in the article 5) testing the hypothesis must be better explained in the methodology
6) testing the hypothesis must be better explained in the methodology
Author Response
Comments 1: Very old bibliography, should focus on recent articles 2021, 2022 and 2023 |
|||
Response 1: Thank you for your comment, a number of old bibliography have been delated such as: -Collis, J.; Hussey, R. Business Research, A Practional Guide for Undergraduate & Postgraduate Students, 3rd ed.; Palgrave Macmillan: New York, 2009. -Langfield-Smith, K. Management Control Systems and Strategy: A Critical Review. Accounting, Organizations and Society 1997, 22, 207-232. -Anderson, S. W.; Young, S. M. The Impact of Contextual and Process Factors on the Evaluation of Activity-Based Costing Systems. Accounting, Organizations & Society 1999, 24, 525-559. -Otley, D. Performance management: a framework for management control systems research. Management Accounting Research 1999, 10, 363-382. -Chang, S.I.; Jan, D. SOX 404-compliant ERP System Internal Control Framework - The Preliminary Outcome. Journal of Business and Policy Research 2010, 5, 282-295. -Klamm, B. K.; Watson, M. W. Sox 404 Reporting Internal Control Weaknesses: A Test of COSO Framework Components and Information Technology. Journal of information systems 2009, 23, 1-23 and a number of recent relative articles have been added, please seepage 15-18 and lines 574, 576, 589,590,591,607,692,722. In addition, the reason for using some old bibliograph is because the internal control standard system was completed in 2010, so a large amount of literature has studied the IC from different perspective around that time (Tao et al. 2023). However, there still a need for more study.
|
|||
Comments 2: The introduction must have a description of the article's sections. the purpose of the article must be better defined. |
|||
Response 2: I agree. I have, accordingly, redefined the purpose of the article and I have tried to clear it in order to emphasize this point, please see page 2 and lines 47-51 & 59-64 & 75-78.
|
|||
4. Additional clarifications |
|||
I have reviewed the introduction, methodology and results. Also, I have added some recent references. Please see the is change, I have highlighted that in the re-submitted files. |

Round 2
Reviewer 3 Report
Comments and Suggestions for Authors
X